# Novel Ge-Based Plasma TFET with High Rectification Efficiency for 2.45 GHz Microwave Wireless Weak Energy Transmission

**DOI:** 10.3390/mi15010117

**Published:** 2024-01-10

**Authors:** Weifeng Liu, Sihan Bi, Jianjun Song

**Affiliations:** 1School of Microelectronics, Xidian University, Xi’an 710071, China; liuweifeng1268@163.com (W.L.); jianjun_79_81@xidian.edu.cn (J.S.); 2State Key Laboratory of Wide-Bandgap Semiconductor Devices and Integrated Technology, Xi’an 710071, China

**Keywords:** microwave wireless energy transfer, weak energy density, TFET

## Abstract

The enhancement of rectification efficiency in 2.45 GHz microwave wireless weak energy transmission systems is centred on rectifier device selection. The overall rectification efficiency of traditional rectifier devices is low in weak energy density situations, failing to fulfil the commercial requirements of this region. The subthreshold swing of the emerging device TFET exceeds 60 mV/dec, which has the advantages of a large open-state current and a small off-state current in the corresponding region of the weak energy density. In view of this, this paper designs a double-gate plasma rectifier TFET with an embedded n^+^ heavily doped layer on the basis of a PNPN-structured TFET, where the device is simulated with the MixedMode module of Silvaco TCAD 2018, the rectification efficiency at −10 dBm is 44.12%, which is 10.61% higher than that of the PNPN-TFET, and the efficiency in the weak energy density region is generally 10% or more than that of commercial HSMS devices, showing obvious rectification advantages.

## 1. Introduction

Microwave wireless energy transfer systems have overcome the limitations of batteries and transmission lines and are now prevalent in many fields including biomedical, aerospace, and radar communication. Additionally, they are minimally affected by environmental factors and terrain, promising a wide range of future applications [1,2,3,4,5,6,7,8]. The main microwave frequency in the Chinese environment is 2.45 GHz, but the energy density is mostly in the region of weak energy density, and the value of the input power that can be received is less than 0 dBm. Devices in this frequency band have an excellent performance in the field of wireless mice and in vitro implantable medical devices. But now the main devices still need to rely on battery power, through the microwave wireless energy transfer system they can take advantage of the energy of the band and can make the energy transfer more efficient and convenient [9,10,11].

Currently, the key rectifier components for microwave wireless energy transfer systems consist of MOSFETs and Schottky diodes. However, MOSFETs face a subthreshold swing limitation of 60 mV/dec, and both components possess low open-state currents in regions with a weak energy density, resulting in less-than-optimal rectification efficiencies [12,13,14]. TFETs exhibit subthreshold swings surpassing the limit of 60 mV/dec, possess low turn-on voltages, and offer the benefits of substantial open-state currents and minimal off-state currents in the area of a weak energy density; different configurations also lead to an increasing TFET performance [15,16]. The corresponding region has the benefit of a high open-state current and low off-state current. Although the PNPN structure of TFET enhances the open-state current of the conventional TFET and has been extensively investigated [17,18,19,20,21,22], the heavily doped source–drain region results in a high leakage current, which impacts the device’s efficiency in weak energy density rectification.

In view of this, this paper introduces the plasma mechanism based on the PNPN-TFET structure. And the weak energy density with an embedded n^+^ heavily doped layer dual-gate plasma rectifier TFET is designed. The structure avoids the heavily doped source–drain process, thus solving the problems of a high leakage current and the difficult process implementation of PNPN-TFET. The device enables new improvements in the TFET rectification performance.

## 2. Novel Ge-Based Plasma TFET Design

### 2.1. Basic Working Mechanism

The TFET operating mechanism is based on the quantum tunnelling mechanism, as shown in Figure 1, which is a schematic diagram of the energy bands of an N-type TFET in the on and off states. When the TFET external gate voltage is 0 V, the drain-side heavily doped N-type semiconductor and the intrinsic channel has a high barrier, and the source-side heavily doped P-type semiconductor and the intrinsic channel barrier width are large, most of the carriers are in the valence band below the electronic tunnelling probability, which is small, and cannot tunnel through the barrier into the channel conduction band, and the device is in the off-state (blue dashed line in Figure 1). When the gate bias voltage is greater than 0 V, the channel region energy band, due to the enhancement of the electric field, is constantly curved downward, the source and channel barrier width increases and decreases with the voltage, the carrier band tunnelling probability increases dramatically, and, at the same time, the channel and drain side of the barrier is also gradually reduced with the voltage, with a large number of carriers from the source tunnelling into the channel and drain side of the movement, and, therefore, the device is open (Figure 1, red solid line).

TFETs, like MOSFETs, also suffer from reverse leakage. However, due to the high tunnelling probability required to turn the device on, there is no current generation by the drift mechanism as there is with MOSFET carriers. The main mechanism for leakage current generation in TFETs is indirect compounding, and the current magnitude is small compared to that of MOSFETs.

### 2.2. Plasma Mechanism

The plasma mechanism was first applied to a charge plasma p-n diode, as shown in Figure 2. Instead of forming N and P regions separately by doping, as in conventional p-n junctions, the diode has two separate metal gates placed on a thin silicon body and making MIS contact with the bulk material through a SiO_2_ dielectric layer, with each gate forming a gold half-contact on either side of the silicon body. If the structural parameters of the device satisfy the basic requirements of the plasma principle, a uniformly distributed plasma of holes and electrons can be induced in the bulk material below the electrodes, each with a different figure of merit, thus forming an undoped PN junction [23,24].

Two conditions are required for the device structure to satisfy the plasma formation. Firstly, the work function of the metal electrode needs to satisfy the inequality:(1)$$\phi_{m,C}<\chi +E_{g}/2$$
(2)$$\phi_{m,A}>\chi +E_{g}/2$$
where $\phi_{m,C}$ is the cathode corresponding to the formation of the N-zone metal electrode work function, and $\phi_{m,A}$ is the anode corresponding to the formation of the P-zone metal electrode work function. χ is the electron affinity potential of the bulk material and $E_{g}$ is the forbidden band width of the bulk material. And for optimum device performance, $\phi_{m,C}$ and $\phi_{m,A}$ must differ by at least 0.5 eV. For the device of Ge as the body material, the cathode area can choose the metal whose work function is less than or equal to 4 eV to be the electrode, such as Hf, La, Tb, Ti, Li, etc., and the anode area can choose the metal whose work function is greater than or equal to 5 eV to be the electrode, such as Co, Pt, Ni, Au, etc., so that the difference between the two electrodes’ work function is enough, and the best performance of the constructed plasma device is achieved.

Secondly, the thickness *t* of the body material should be less than the Debye length *L*_D_:(3)$$t<L_{D}=\sqrt{\left(\varepsilon \cdot V_{T})/(qN\right)}$$
where *V*_T_ is the thermal voltage, *N* is the carrier concentration of the bulk material, and ε is the dielectric constant of the bulk material. The charge under the two electrodes is mainly determined by the carriers, and the depletion charge can be neglected independently of the doping concentration in the bulk material. For Ge-based TFETs, the body material thickness is around 10 nm to achieve a better performance.

Therefore, if the plasma mechanism is introduced into the PNPN-TFET, the source and drain regions of the device do not need to be doped, and the same device performance can be achieved by simply designing suitable metal electrodes, which not only avoids the complex process of heavy doping, but also solves the leakage problem caused by heavy doping.

### 2.3. Device Structure Design and Parameter Optimisation Simulation

Based on the study in the previous section, this section introduces the plasma mechanism into PNPN TFET in order to design a new TFET with a higher rectification performance at a low energy density. The selection of metal electrodes used to sense the plasma in the source–drain, the selection of single-gate and double-gate structures, and the selection of the full-envelope and half-envelope structures of the metal electrodes all affect the final performance of the devices. Therefore, in this section, the design of an embedded n^+^ heavily doped layer dual-gate plasma rectifier TFET is designed from the above influencing factors. Based on this, the rectifier circuit is built to evaluate its rectification performance.

The Silvaco TCAD simulation was set up with the bbt.nonlocal band-tunnelling model to simulate the particle tunnelling scenario, and a fine quantum mesh was set up at the tunnelling junctions of the device to make the iterative calculation of the tunnelling probability more accurate. Conmob and fldmob mobility models were also set up to simulate the motion of the charge carriers in the electric field. The bgn energy band narrowing model is set up to simulate the change in the effective width of the energy band. For the electron–hole confinement effect, SRH indirect confinement and Auger-Russian confinement are used to accurately describe the effect of the confinement current on the device characteristics.

The plasma structure is introduced into the PNPN TFET, and the structures of the reference PNPN TFET and the designed plasma TFET with embedded n^+^ heavily doped layers are shown in Figure 3. Figure 3a gives the structure and doping concentration of the four doping regions that make up the PNPN TFETs. The structural parameters of the plasma TFET are given in Figure 3b, and the key device structure parameters required for the simulation are given in Table 1. The device source, channel, and drain are all made of Ge body materials, without the need for source–drain redoping, and the theoretical body material thickness is set to 10 nm from the previous section to ensure that the plasma mechanism induces uniform electrons and holes. The parameters of the embedded n^+^ heavily doped layer are based on the PNPN TFET with a width of 3 nm and n^+^ doping $\left(N_{D}=5\times 10^{19}\;cm^{-3}\right)$, and the presence of the embedded layer maintains the advantage of a high open-state current in the open state of the device.

Based on the theory in the previous section, the selection of the source–drain electrode metal has an important impact on the performance of the plasma device, so under the premise that the source–drain electrode metal figure of merit satisfies inequalities (1) and (2), the source–drain metal electrodes are selected for simulation under different figures of merit. Figure 4 shows the transfer characteristic curves of plasma TFET under different source metal electrodes; with the decrease of the electrode power function, the transfer characteristic curve is steeper, the device subthreshold swing is better, and it is more conducive to the weak energy density rectification, so the Co material with a metal work function of 5 eV is selected as the source metal electrode. Figure 5 shows the transfer characteristic curves of plasma TFETs with different drain metal electrodes, and it can be seen from the local zoomed-in figure that, as the metal work function decreases, the device current fluctuates drastically at a negative voltage, which is not conducive to the smoothness of the rectified output current, and, therefore, the Hf material with a figure of merit of 3.9 eV is selected as the drain metal electrode.

The structure of the source–drain metal electrodes of plasma TFETs has an important impact on the device performance, the fully enveloped structure of the electrodes provides the more comprehensive plasma sensing of the source–drain electrons and holes, and a uniform plasma charge is induced at both the top and bottom of the device, thus contributing to the tunnelling current. Figure 6a shows a double-gate plasma TFET with a half-envelope structure at the source–drain electrode, and Figure 6b shows the final design of a double-gate plasma TFET with an embedded n^+^ heavily doped layer and a full-envelope structure at the source–drain electrode. The key parameters of the device are consistent with the base plasma TFET designed in Figure 3. Various parameters of the bottom gate and source–drain electrodes as well as the oxide layer are consistent with the top one, and the overall structure is symmetrical.

Figure 7 shows the electron and hole concentration distributions from the source region to the drain region under Figure 7a the top gate and Figure 7b the bottom gate for the fully enclosed source–drain electrode (DE) and semi-enclosed source–drain electrode (SE) TFETs. There is no significant difference in the induced electron and hole concentrations between the two devices due to the same metal source–drain electrode structure at both ends of the top gate. At the bottom gate, the hole concentration in the source P region and the electron concentration in the drain N region of the fully encapsulated source–drain electrode device, shown as the solid line in the figure, are higher than those of the device with the half encapsulated source–drain electrode structure, shown as the dashed line, so it can be seen that the fully encapsulated electrode structure can sense a higher concentration of electrons and holes and the plasma sensing effect is better.

Figure 8 shows the transfer characteristics of the fully enclosed source–drain electrode and half-enclosed source–drain electrode TFETs in the V_DS_ from 0.2 V to 0.6 V. As the applied voltage increases, the open-state current of the fully enclosed source–drain electrode structure of the TFETs is greater than that of the half-enclosed source–drain electrode structure. As can be seen from the above, the fully enclosed source–drain TFET has a better performance than the half-enclosed source–drain TFET, and the device has a higher output power by driving a high current in the low energy density region, which is conducive to improving the rectification efficiency. Therefore, the fully enclosed source–drain electrode-embedded n^+^ heavily doped layer plasma TFET of Figure 6b is identified as the final rectifier device, which is referred to as a novel TFET for the convenience of the following presentation, and its performance is compared with that of the PNPN TFET and evaluated by a circuit simulation.

Figure 9 shows the group diagram of the designed novel plasma TFET compared with the PNPN-TFET. Figure 9a shows the open-state energy band diagrams of the two devices, and the width of the tunnelling barriers of the two devices at the tunnelling junction are similar, which shows that the new plasma TFET maintains the tunnelling advantage of the PNPN-TFET structure. Figure 9b shows the off-state energy band diagrams of the two devices, and the new plasmonic TFET has a higher barrier at the drain end, which can well suppress the reverse carrier motion under reverse bias. Figure 9c depicts the variation of the electron concentration along the transverse cross section of the device. In the on-state, the electron concentration on the source side is low and the electron concentration on the drain side is maximum because the source is a P-region and the drain is an N-region. The electron concentration distribution at the drain side is similar for both devices and is in the same order of magnitude. Figure 9d depicts the variation of the hole concentration along the transverse intercept of the device. In the on-state, the hole concentration at the source side is maximum and the hole concentration at the drain side is lower. And the two devices have similar hole distributions at the source side, which indicates that the plasma sensing is more effective. Figure 9e depicts the variation of the potential along the transverse intercept of the device. The potential is lower at the source side of the device and maximum at the drain side, and the potential distribution is similar in both devices. Figure 9f depicts the variation of the electric field along the transverse truncation of the device. The electric field at the tunnelling junction of the new plasmonic TFET is still somewhat inferior compared to the heavily doped PNPN structure, but the electric field at the drain side is also smaller than that of the PNPN-TFET, which is favourable for suppressing the reverse leakage. Figure 9g shows the BTBT (band-to-band tunnelling) rate distributions of the two devices. The BTBT model integrates the tunnelling of the carriers across the depletion layer between the source and channel interface. The two devices have similar peak tunnelling probabilities at the tunnelling junction and have a similar performance.

Figure 10a shows the transfer characteristics of both devices in the low energy density region corresponding to a low voltage. The on-state current of the novel plasma TFET is slightly higher than that of the PNPN TFET in the double-gate structure, while the off-state current decreases by almost two orders of magnitude in the reverse current due to the avoidance of a heavily doped source–drain. Figure 10b shows the current switching ratios of the two devices at different bias voltages. The novel plasma TFET has a better switching performance due to the reduction of reverse leakage, and the sub-peak caused by reverse leakage in the input power can be significantly reduced when applied to rectifier circuits, thus improving the rectification performance of the device.

Figure 11a shows the plot of transconductance (gm1) versus gate voltage for both devices. Figure 11b shows the second order transconductance (gm2) and Figure 11c shows the third order transconductance. In 2.45 GHz high-frequency circuits, the high frequency of the signal and the short response time of the circuit require the device to have a fast response speed. The higher-order transconductance of the new plasma TFET is higher than that of the PNPN TFET in the low energy density region, which can be applied to circuits with a higher gain and faster response time for the better reception of high-frequency signals.

In summary, the new plasma TFET finally designed in this subsection, i.e., the double-gate plasma TFET with embedded n^+^ heavily doped layers in the all-around source–drain electrode, retains the excellent high open-state current of the PNPN TFET and greatly reduces the reverse leakage, and it will have a better rectification performance in the region of a weak energy density at a high frequency of 2.45 GHz. A rectification circuit will be constructed in the next subsection to make the final quantitative evaluation.

## 3. Rectifier Circuit Design and Simulation

### 3.1. Half-Wave Rectifier Circuit Construction

For the new plasma TFET designed in the previous subsection, a rectifier circuit is built, as shown in Figure 12. The device electrode is a dual-gate structure, with the dual-gate together with the drain connected as the output, the source as the input, forming the rectifier connection, while the gate needs to be short-circuited in the mixed-mode module instruction to properly connect the dual-gate. The input power supply of the circuit is a 2.45 GHz AC voltage source, the internal resistance of the power supply *R* is 50 Ω, and *R*1 is a load resistor of 20 kΩ which plays a role in stabilising the output voltage. The output of the rectifier circuit will be unstable if the resistance is too small, and too much energy will be wasted by the circuit if the resistance is too large. *C*1 is a load capacitor of 1 pF which plays a role in filtering the output voltage to make the output voltage smoother.

### 3.2. Transient Simulation

The constructed TFET half-wave rectifier circuit is analysed by a transient simulation using the Silvaco MixedMode module to quantify its rectification performance at a weak energy density. Rectification efficiency is the core indicator of the device rectification performance, the higher the rectification efficiency, the more energy can be utilised, which is calculated as shown in Equation (4). As shown in Equation (5), to calculate the rectification efficiency of a cycle it is necessary to integrate the output and input power of this cycle and compare it.
(4)$$\eta =\frac{P_{out}}{P_{in}}\times 100\%=\frac{I_{out}\times V_{out}}{I_{in}\times V_{in}}\times 100\%$$
(5)$$\eta =\frac{\frac{1}{T}\int_{0}^{T}I_{out}\times V_{out}}{\frac{1}{T}\int_{0}^{T}I_{in}\times V_{in}}\times 100\%$$

Figure 13 shows the output current–voltage comparison between the novel plasma TFET and the PNPN-TFET after a transient simulation. At this time, the peak input AC voltage was 0.1 V, extracted as a 30~31 ns stable waveform, the peak output voltage of the new plasma TFET increased by 0.017 V, and the output current increased by 0.82 μA. The output current and voltage are presented as a stable and regular DC waveform, in which it can be seen that the new plasma TFET retains an excellent rectification performance, and the output power can be maintained at a high value.

Figure 14 shows a plot of the reverse peak of the input current of the new plasma TFET versus the PNPN-TFET. The input to the rectifier circuit is an AC signal, and the rectifier device will turn off completely in the ideal case when reverse bias is applied, but due to the presence of the reverse leakage current, the reverse current peak, as shown in the figure, will be presented, which will have an effect on the output pulsating DC. The plasma TFET has a lower reverse leakage current, which reduces the current reverse peak by a factor of 104.64 compared to the PNPN-TFET, which facilitates the stabilisation of the output DC signal and reduces the effect of the negative half week of the inlet and outlet power on the overall rectification efficiency of the device, which improves the device’s rectification efficiency.

Figure 15 shows the comparison curve of the rectification efficiency of the new plasma TFET with the PNPN-TFET, while the commercial rectifier devices HSMS-2850 and HSMS-2860 from Agilent (Santa Clara, CA, USA) were selected for comparison [25]. And we also selected TSMC’s commercial MOS as a comparison [26]. The new plasma TFET reaches a 67.12% rectification peak at a weak energy density, has a rectification efficiency of 20.65% at −15 dBm, 6.73% higher than that of the PNPN-TFET, has a rectification efficiency of 44.12% at −10 dBm, 10.61% higher than that of the PNPN-TFET, and is more gentle in the rectification efficiency drop. Agilent’s two devices can have considerable rectification efficiency in the weak energy density, but they are not yet able to reach the weak energy density’s dedicated commercial needs. Compared with commercial rectifier devices in a weak energy density below 0 dBm rectification efficiency, the two TFETs designed in this paper have obvious advantages.

## 4. Conclusions

In this paper, we propose and design a new Ge-based plasma TFET with high rectification efficiency for 2.45 GHz microwave wireless weak energy transmissions, which solves the problems of the increased reverse leakage current and high process complexity of PNPN-TFETs, with a rectification efficiency of 20.65% at −15 dBm, which is 6.73% higher than that of PNPN-TFETs, and a rectification efficiency of 44.12% at −10 dBm, which is 10.61% higher than that of PNPN-TFETs. At −10 dBm, the rectification efficiency is 44.12%, which is 10.61% higher than that of PNPN-TFET. The peak efficiency of the device is as high as 67.12%, and the efficiency in the weak energy density region is generally higher than that of commercial devices by more than 10%, which fully reflects the advantages of the device in the weak energy wireless transmission, and fills the research gap of TFET in the field of weak-energy microwave wireless energy transmission.

## Figures and Tables

**Figure 1 micromachines-15-00117-f001:**
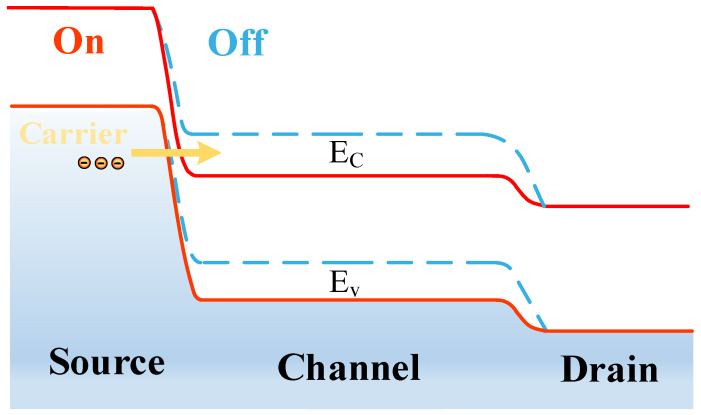
Ge-based N-type TFET on and off state energy band diagrams.

**Figure 2 micromachines-15-00117-f002:**
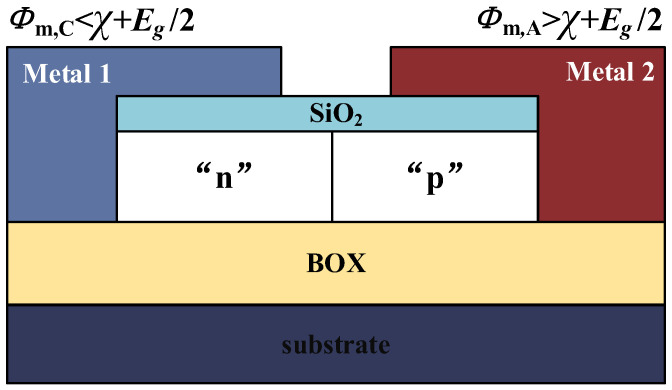
Plasma p-n diode structure.

**Figure 3 micromachines-15-00117-f003:**
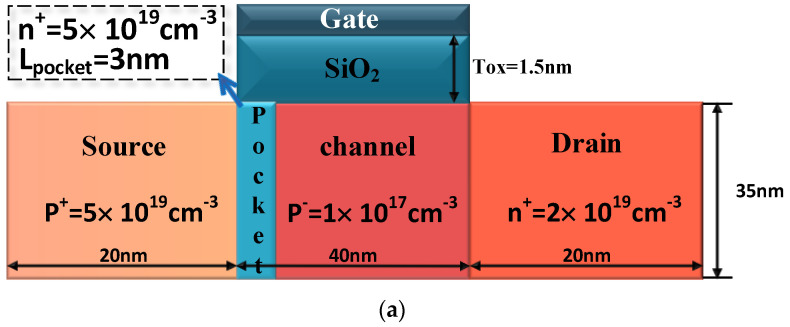
Structure of (**a**) PNPN TFET; (**b**) plasma TFET with embedded n^+^ heavily doped layers.

**Figure 4 micromachines-15-00117-f004:**
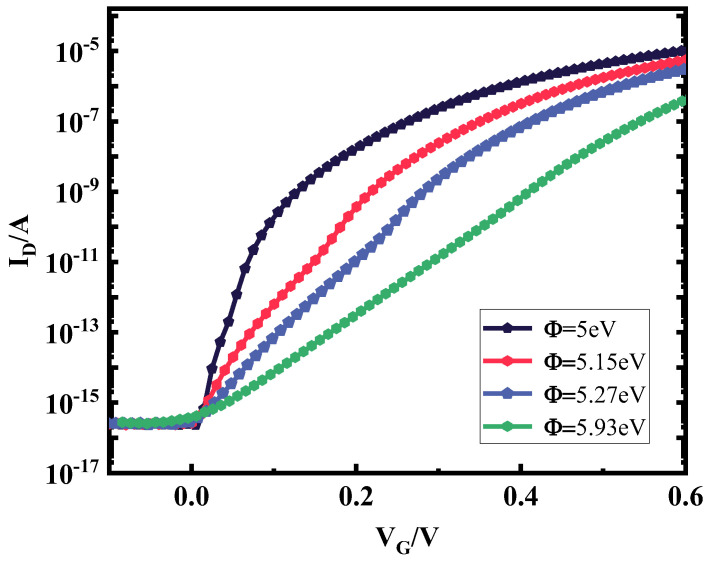
Plot of plasma TFET transfer characteristics at different source metal electrodes.

**Figure 5 micromachines-15-00117-f005:**
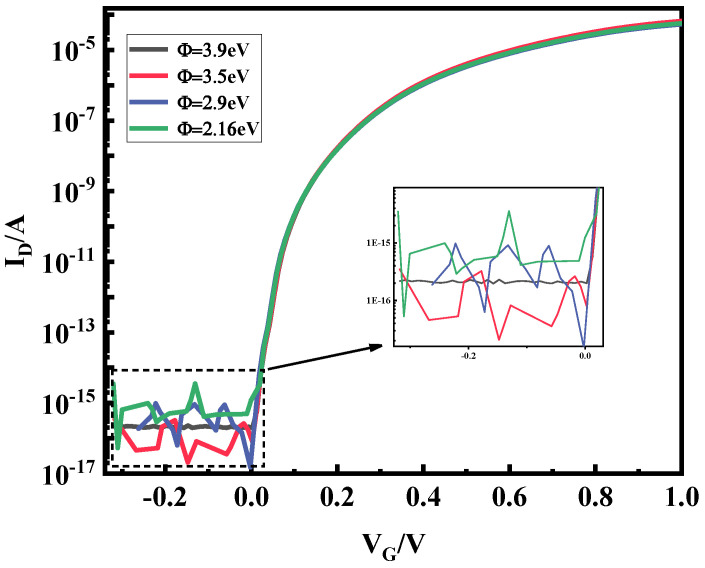
Plot of plasma TFET transfer characteristics with different drain metal electrodes.

**Figure 6 micromachines-15-00117-f006:**
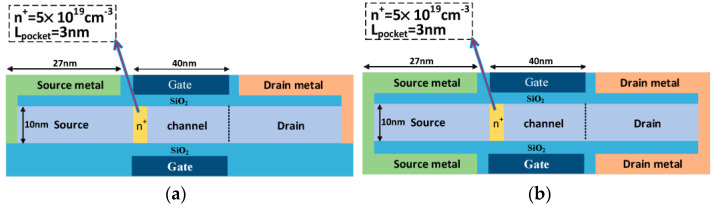
Embedded n^+^ heavily doped layer plasma TFET. (**a**) Dual-gate semi-surrounded source–drain electrode structure; (**b**) Dual-gate fully surrounded source–drain electrode structure.

**Figure 7 micromachines-15-00117-f007:**
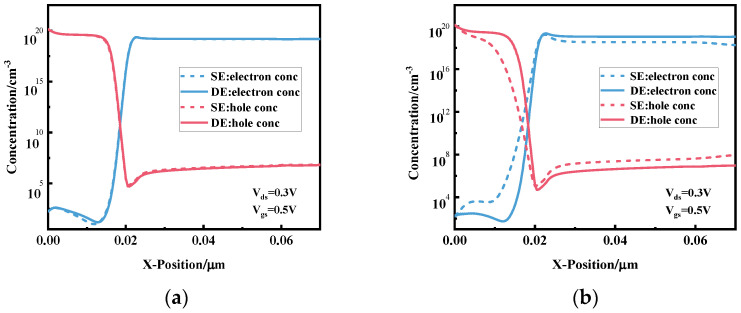
Distribution of electron hole concentration in the vicinity of (**a**) top gate; (**b**) bottom gate for fully surrounded and half-surrounded source–drain electrode TFETs.

**Figure 8 micromachines-15-00117-f008:**
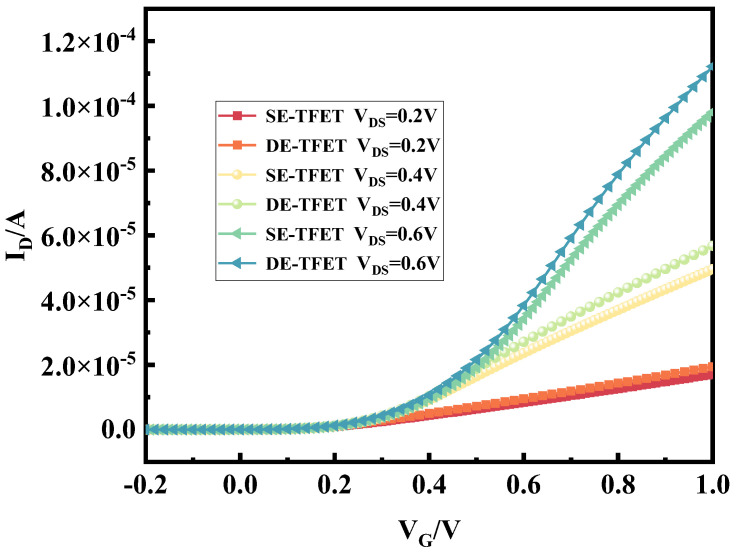
Full-envelope and half-envelope source–drain pole TFET transfer characteristic curves.

**Figure 9 micromachines-15-00117-f009:**
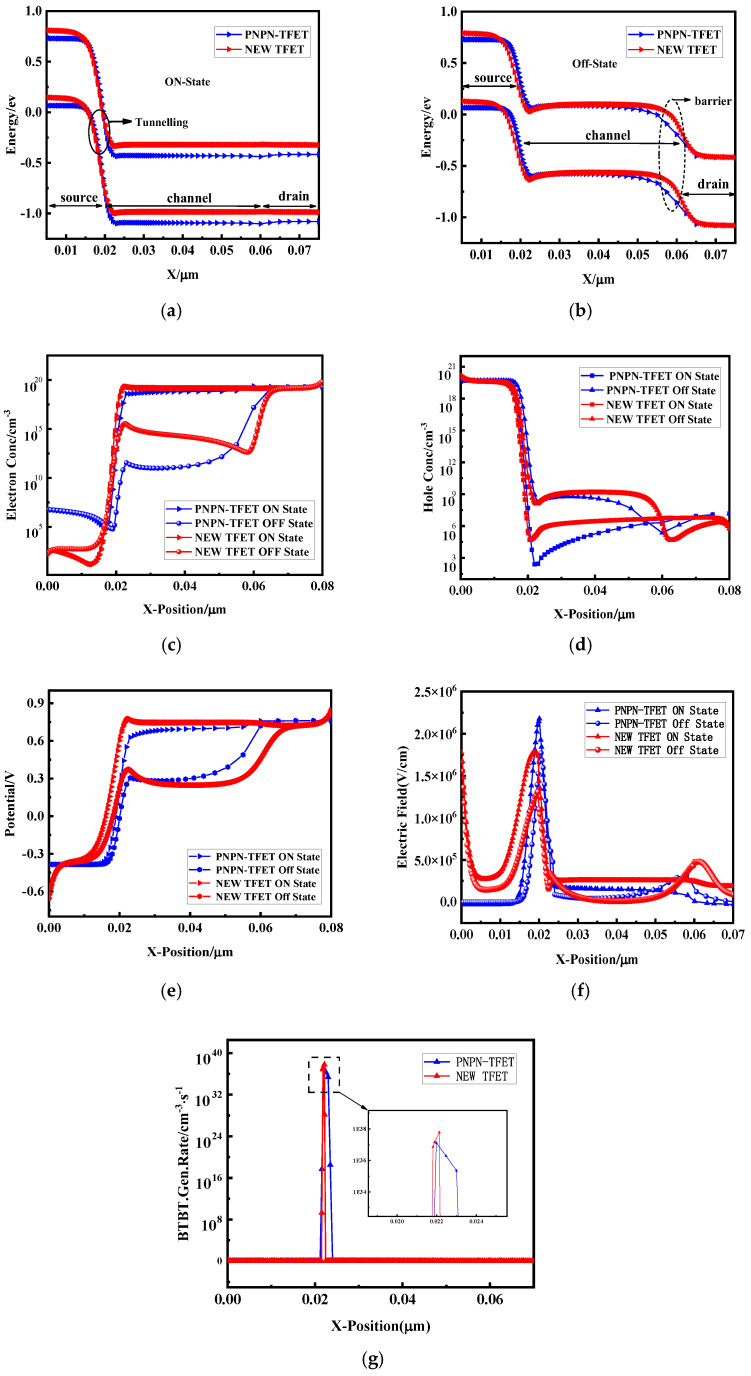
Comparison of new plasma TFET with PNPN-TFET at one-side gate, (**a**) open-state energy band; (**b**) closed-state energy band; (**c**) electron concentration distribution; (**d**) hole concentration distribution; (**e**) potential distribution; (**f**) electric field distribution; (**g**) BTBT rate distribution.

**Figure 10 micromachines-15-00117-f010:**
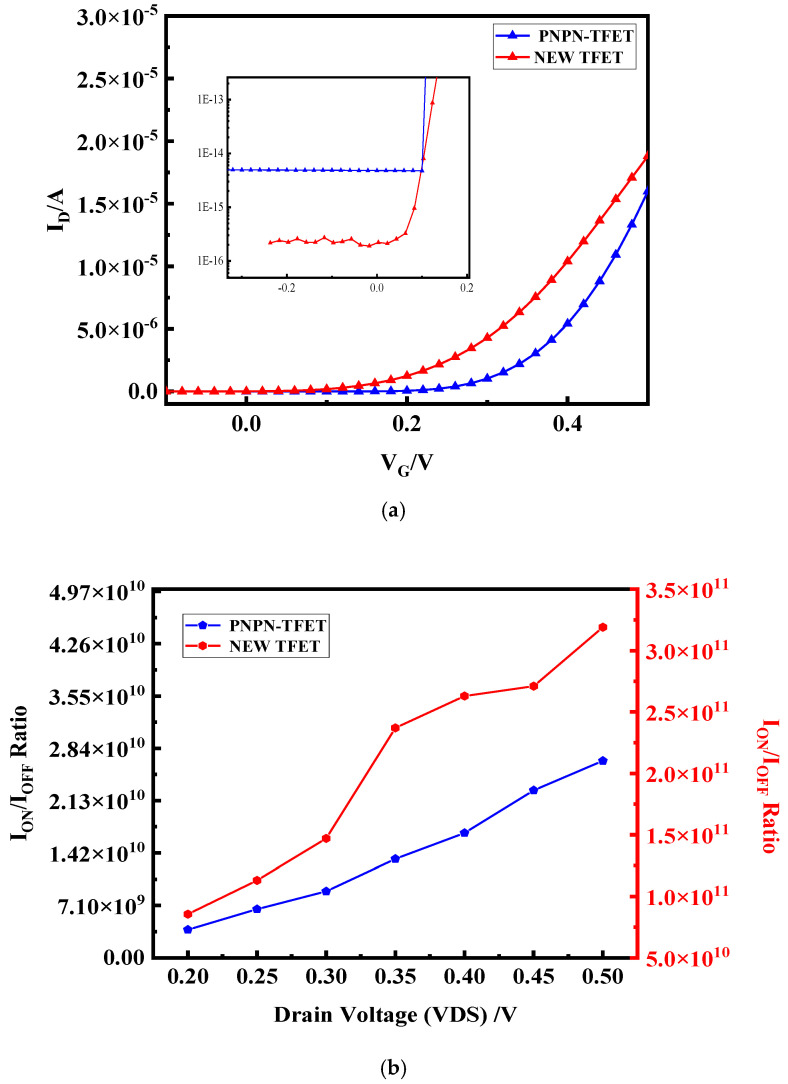
Comparison of new plasma TFET with PNPN-TFET, (**a**) transfer characteristic curve; (**b**) current switching ratio.

**Figure 11 micromachines-15-00117-f011:**
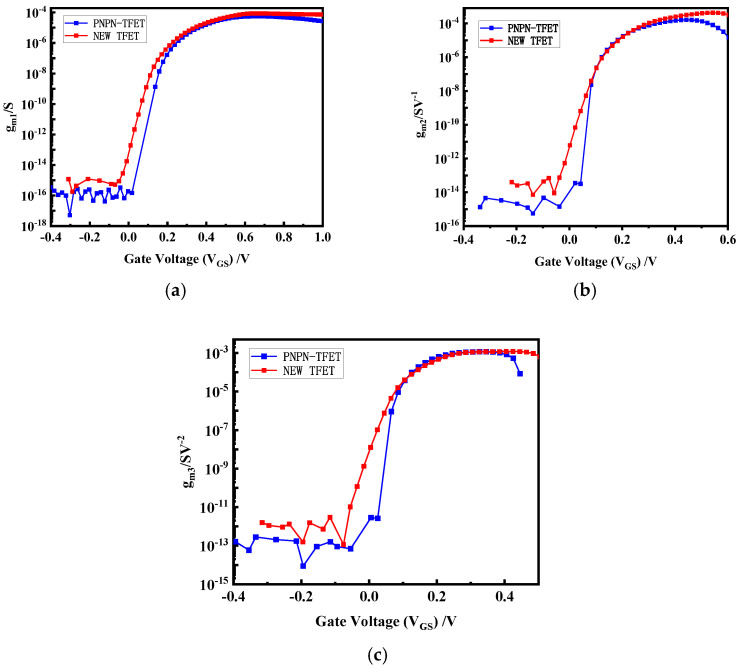
Comparison of new plasma TFET with PNPN-TFET, (**a**) transconductance; (**b**) second-order transconductance; (**c**) third-order transconductance plots.

**Figure 12 micromachines-15-00117-f012:**
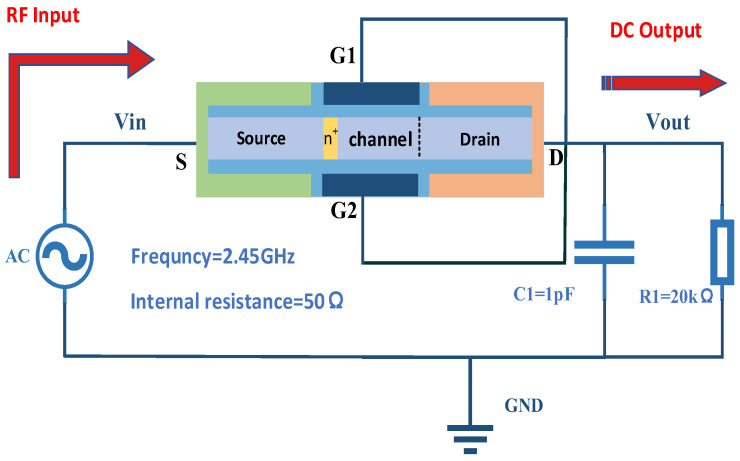
New Plasma TFET Rectifier Circuit Diagrams.

**Figure 13 micromachines-15-00117-f013:**
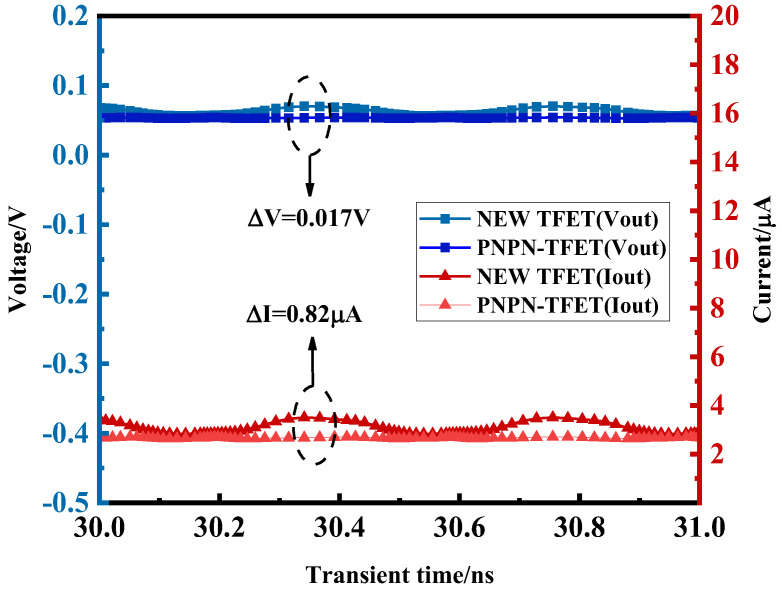
Output current–voltage plots for transient simulation of new plasma TFET and PNPN-TFET.

**Figure 14 micromachines-15-00117-f014:**
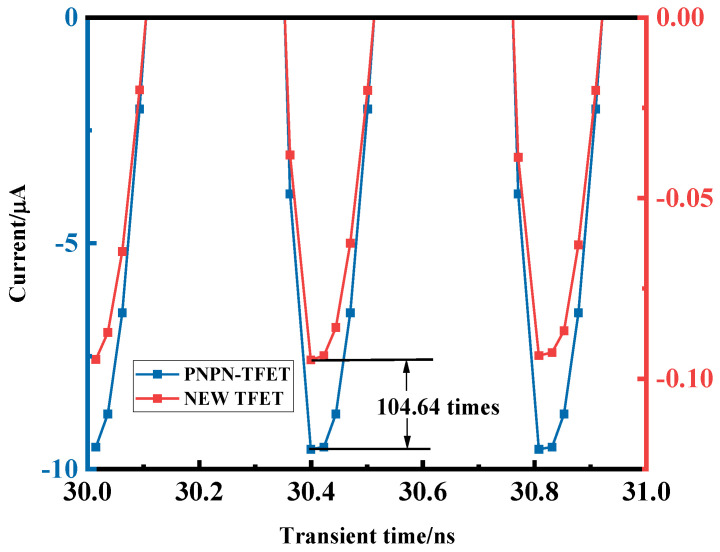
Plot of new plasma TFET versus PNPN-TFET with reverse peak input current.

**Figure 15 micromachines-15-00117-f015:**
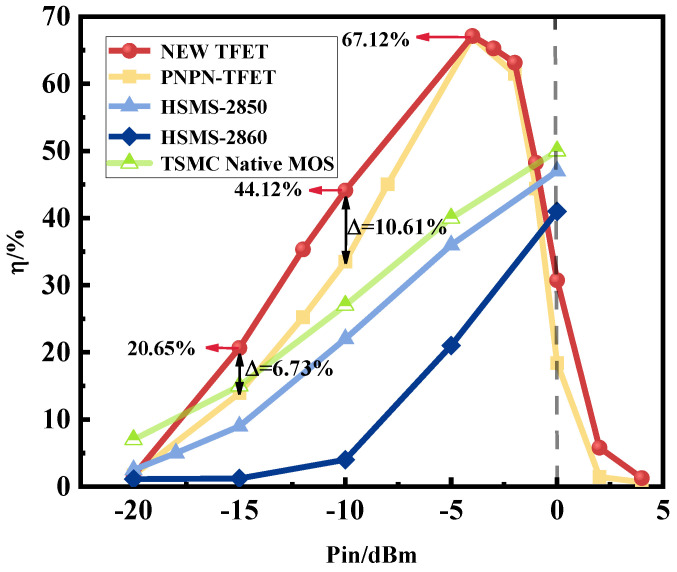
Comparative Curves of Rectification Efficiency of New Plasma TFET.

**Table 1 micromachines-15-00117-t001:** Device Simulation Structure Parameter Table.

**Parameter**	**Value**
Ge body material thickness (nm)	10
Source metal length (nm)	27
Drain metal length (nm)	27
Gate length (nm)	40
Oxide layer thickness (nm)	1.5
Source gate pitch (nm)	3
Drain gate pitch (nm)	3
Source metal work function (eV)	5
Drain metal work function (eV)	3.9

## Data Availability

The data presented in this study are available on request from the corresponding author. The data are not publicly available due to some data which are to be used in a particular author’s degree thesis, and it is important to prevent data theft.

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
