# Peer review of "Novel Ge-Based Plasma TFET with High Rectification Efficiency for 2.45 GHz Microwave Wireless Weak Energy Transmission"

_micromachines, 2024, doi:10.3390/mi15010117_

Round 1
Reviewer 1 Report
Comments and Suggestions for Authors
In this manuscript, a double-gated plasma TFET has been proposed which is shown to be a better rectifier in the low voltage regime when compared to the traditional PNPN-structured TFET. The manuscript is clearly structured and logically rigorous. The main message is well conveyed.
Please find a few questions/comments below:
1. You have mentioned the term “weak energy system” quite a few times since the device you designed operates best in this regime when compared to the conventional devices. Could you please define “weak energy system” in the beginning? For instance, what energy/power/voltage range one can consider as weak energy? Also, clearly mention a few applications with references where this regime is important.
2. Although the conclusion is clearly stated and the research findings are linked with the research question, it would be nice to see the operating range and conditions of this device here, along with some potential application. Also, I understand why the authors focus on the 2.45 ISM radio band. However, it would be great to have one or two sentences on how the device would perform for other frequency band.
3. The authors should provide more information on the TCAD simulation. For example, the doping information (especially for the PNPN-TFET), mobility models/values, any other device physics model that has been used in Silvaco. In addition, for PNPN-TFET, more references are required for the readers to refer for its structure and performance.
4. On page 4 and 5, what do you mean by metal figure-of-merit (line 144) and metal power function (line 141)? Do you mean metal work function?
5. Fig 9g is not "tunneling probability". Please use correct figure caption and maybe discuss how the term you are plotting is related to the tunneling probability.
6. For a smoother read, maybe use subscripts to represent your variables. For example, VDS, ION, IOFF etc. are not generally used. Instead, VDS, ION, IOFF are most commonly used across literature and textbooks.
Comments on the Quality of English LanguageThe manuscript is difficult to read at times. Some sentences are long and convoluted. I would suggest breaking down of the long sentences into short and simple ones. The following parts may be simplified:
“This paper is based on the PNPN-TFET structure and introduces a plasma mechanism to design a dual-gate plasma rectified TFET with an embedded n+ heavily doped layer for low energy density. This structure avoids the heavily doped source-drain process, solving the issues of high leakage current and challenging process realisation faced by PNPN-TFETs, and enhancing the rectification performance of TFETs.”
“Based on the study of the plasma mechanism in the previous section, this section introduces the plasma mechanism into PNPN TFETs in order to design new TFETs with higher rectification performance at a low energy density. In designing new plasma rectified TFETs, the selection of metal electrodes used to sense the plasma in the source-drain, the selection of single-gate and double-gate structures, and the selection of full-envelope and half-envelope structures of the metal electrodes all influence the final device performance. Therefore, in this section, we design the double-gate plasma rectifier TFET with embedded n+ redoped layer from the above influencing factors and build a rectifier circuit to evaluate its rectification performance.”
Reviewer 2 Report
Comments and Suggestions for Authors
Wireless transmission is a very popular area of research. In this manuscript, the authors designed a double-gate plasma rectifier TFET with embedded n+ heavily doped layer on the basis of a PNPN-structured TFET. The device shows excellent application prospects in the RF rectification field. Overall, this paper is interesting and the idea is novel. And I suggest the following revision before publishing on micromachines.
1. Please standardize the connectors between physical quantities and units on one side of the axes in the picture, for example, 'VG/V' and 'X-Position(μm)'.
2. Please improve your English writing skills and you can seek help from native English speakers.
3. For the analysis of the rectification efficiency of the new TFETs in Figure 15, is it possible to add rectified commercial MOSFETs for rectification efficiency comparison, making the high rectification performance of the devices more convincing?
Comments on the Quality of English LanguagePlease pay attention to the use of "the"
Round 2
Reviewer 1 Report
Comments and Suggestions for Authors
The authors have addressed the comments. I recommend the manuscript for publication.